# The impact of digital economy on high-quality economic development: Research based on the consumption expansion

**Xiaoxuan Li** [1]*, **Qi Wu** [2]

**1** School of Economics, Fuyang Normal University (FYNU), Fuyang, Anhui, China, **2** School of Physics and Electronic Engineering, Fuyang Normal University (FYNU), Fuyang Anhui, China

\* 201607016@fynu.edu.cn

**Data Availability Statement:** All relevant data are within the manuscript and its Supporting Information files.

**Funding:** This research was funded by Natural Science Foundation of Anhui Province (No. 1908085QG305), Scientific Research Foundation

## Abstract

Studies have found that the digital economy plays a positive role in promoting high-quality economic development. Meanwhile, the digital industrialization and industrial digitalization have given rise to new demands and supply modes of consumption. It is necessary to analyze the role of consumption expansion in the impact of digital economy on high-quality economic development. Based on Chinese provincial panel data, we first applied the entropy weight method to construct the digital economy index and the high-quality economic development index. And on this basis, it was verified that the digital economy can positively promote the high-quality development of the inverted U-shaped structural economy. Then we separately used the mediation and the threshold effect models to analyze the role of consumption expansion in empowering high-quality development in the digital economy. Regional heterogeneity was further taken into account. The results dedicate that consumer demand and the digital economy have a partial or complete mediating effect. The promotion of high-quality development by the digital economy can be affected by the threshold of consumption expansion, which is manifested in the marginal incremental effect due to the growth of consumption supply. On the contrary, the growth of consumer demand has led to the inverted U-shape of the digital economy to promote high-quality economic development. In the heterogeneity analysis, the threshold effect also varied greatly. This research enriches the theoretical achievements and reveals the impact of consumption expansion on the digital economy affecting the high-quality development, which may provide certain reference for other countries and regions.

## Introduction

Nowadays Internet information technology is being rapidly popularized and applied with the march of economic globalization. The digital economy, characterized by digitalization, networking and intelligence, is rising rapidly and has become the main driving force for comprehensive economic and social innovation and change [1]. The digital economy has penetrated and changed patterns of social life and production activities. In the meantime, China has

of Education Department of Anhui Province of China (No. SK2020A0341). The funders had no role in study design, data collection and analysis, decision to publish, or preparation of the manuscript.

**Competing interests:** The authors have declared that no competing interests exist.

adhered to the new development concept of "innovation, coordination, green, openness and sharing" [2]. Many factors including global climate change affect economic development and economic quality [3]. Achieving high-quality economic development has become one of the priorities of China's economic work. Therefore, the key issue that needs to be addressed urgently is to identify the entry point and force point for achieving high-quality development and promoting common prosperity, making use of resource endowments, development conditions, and comparative advantages.

The characteristics of the digital economy include data support, innovation integration, and open sharing [4], which coincide with the connotation of high-quality development. This provides new ideas to achieve high-quality economic development for China. At present, the most urgent topic is how to use the digital economy as a driving force to promote the transformation and upgrading of economic structure and sustainable high-quality economic development. Based on different research perspectives and methods, some scholars believe that there is a significant positive correlation between the digital economy and high-quality economic development [5–7]. However, this positive relationship has a more complex influence path, which often appears as nonlinear features [8]. Based on the perspective of entrepreneurial activity, when studying the quality of economic development in the EU, it was found that there is a significant difference between the development of digital economy and the relationship between entrepreneurial activity and economic growth [9]. When high growth is expected, the more digitalized the regions are, the greater the contribution of entrepreneurial activity to economic growth. When the average growth is expected, the more entrepreneurial activity plays a leading role in economic growth in regions with lower levels of digitalization [10]. From the perspective of industrial upgrading, it was found that the digital economy can not only directly promote high-quality economic development, but also indirectly promote the high-quality development of the economy through industrial upgrading [11–13]. Based on China's provincial panel data, scholars have verified the influence mechanism and role of digital economy in high-quality development, and the results show that there is a positive promotion effect between digital economy and high-quality development [14]. The impact of the digital economy on total factor productivity is U-shaped, which indirectly promotes high-quality economic development [15,16].

In fact, consumption can also have a great impact on the path of digital economy affecting the high-quality development of the economy [17]. At the micro level, economies of scale and of scope brought by digital technology can better match supply and demand, thus realizing high-quality economic growth [18]. First of all, the consumption concept of the public has gradually changed under the influence of the digital economy [19]. With the promotion of digital economy technology, social consumption tendencies have changed, and many studies have shown that the digital economy can effectively promote consumption [20–22]. On the basis of tapping the consumption potential of the digital economy [23], comprehensively promoting the transformation of social consumption [24], and reshaping the characteristics of social consumption [25], the activity of market transactions will be significantly increased [26]. There is also a strong correlation between the digital economy and consumer behavior, which is reflected in the transformation of online transactions, consumer price sensitivity, and the protection of consumers' personal information [27]. Secondly, while expanding domestic demand, it is also necessary to pay attention to innovation-driven, providing high-quality consumption supply to meet social demand, and gradually improving the market supply system [28]. The development of digital technology can promote the optimization and upgrading of industrial structure and bring positive effects to industrial adjustment and social consumption transformation and upgrading [29]. If the concept of digital economy technology can be fully introduced into economic development, it will effectively stimulate residents' desire for

consumption, improve consumption supply capacity and quality, and thus provide impetus for sustainable economic development [30]. Enterprise digital transformation will conduce to enhancing the innovation consciousness of enterprises and laying a solid foundation for the upgrading of social consumption structure [31]. The application of digital economy technology in traditional industries can lead to more flexible manufacturing, which can meet the diversified needs of consumers more effectively. Digitalization of traditional enterprises can also enrich customer service channels by making better use of special manufacturing and smart manufacturing [32].

From the existing researches, we found that consumption expansion is rarely taken into account in the impact of the digital economy on high-quality development. Will the consumption expansion affect digital economy's impact on promoting high-quality development? And if it does, what is the relationship between them? What about the impact pattern and effect? These questions are all critical, but the conclusions are yet unclear. Clarifying the role of consumption expansion will help to better achieve high-quality economic development. Based on the above analysis, this paper intends to study the impact of digital economy on high-quality economic development, and the effect of consumption expansion in the influence path of digital economy empowering high-quality economic development. This also has certain guiding significance for other countries to promote the balance of market supply and demand, and improve the coordinated development efficiency of digital economy and high-quality economic development.

## Theoretical analysis

### Digital economy and high-quality economic development

With digital technology as the core driving force, the digital economy can directly promote the dynamic, efficiency and quality transformation of economic and social development through information technologies such as artificial intelligence, big data, and cloud computing [33]. Regarding the research on the relationship between digital economy and high-quality economic development, scholars generally believe that the digital economy can positively promote high-quality economic development [34]. The theory of high-quality economic development is rich in connotation, and it is highly consistent with the five new development concepts proposed by China [35]. Industries related to the digital economy are characterized by high knowledge density and strong innovation attributes, and their development and utilization of new technologies, new methods and new products provide a solid digital foundation for high-quality economic development [36]. The digital economy can reduce the cost of key factors, exert the network radiation effect, improve the allocation efficiency of production factors, and ultimately improve the efficiency of regional resource allocation. We should give full play to the advantages of the digital economy in various regions, and use the spillover effect of the digital economy to drive the long-term development of less developed regions, so as to narrow the development gap between regions [37]. The digital economy can effectively transform the input and output mode of traditional production factors, and enhance the ability of resource integration and ecological environment governance. This helps to improve the combined efficiency of green production factors, which is reflected in the improvement of green total factor productivity [38]. With the in-depth development of digital technology, cross-border e-commerce, as an emerging force for expanding opening up, can strengthen economic and trade exchanges between countries. This is helpful to promoting regional economic integration [39]. The development of the digital economy has greatly improved the efficiency of matching supply and demand of the economy, expanded effective demand, and ensured a sufficient and diverse material base [40]. Based on this, we proposed hypothesis 1:

H1: The digital economy can positively promote high-quality economic development.

## The nonlinear influence

Compared with the traditional economy, the digital economy is the manifestation of the industrialization and marketization of the information technology revolution. Its development conforms to Metcalfe's law, Moore's law and Davido's law [41]. This determines that the digital economy has the characteristics of increasing marginal returns and diminishing marginal costs. In the initial stage of the digital economy, the construction of digital infrastructure is incomplete, and the digital dividend cannot be effectively released, which converge to the limited promotion of high-quality economic development [42]. With the iterative upgrading of digital technology and the continuous development of digital industry, the transaction costs between economic entities and the marginal cost of digital technology Research and Development (R&D) continue to decline, and the scale returns of market entities increase. The economic benefits obtained from the digital economy increase exponentially, and the role of empowering the high-quality development will increase rapidly. With the continuous development of the digital economy and upgrading of the digital industrial structure, the potential of the digital economy will be gradually released. At this point, the force driving high-quality economic development will show the law of diminishing marginal utility. Accordingly, hypothesis 2 was proposed as:

H2: The influence of digital economy on high-quality economic development is non-linear.

## The impact of consumption expansion

Both the development of digital economy and consumption growth will impact on high-quality economic development. This impact path includes both consumer demand and consumption supply. For one thing, users can utilize the digital information platform to obtain commodity information in a more timely and effective manner, while reducing the cost of commodity information search and information asymmetry [43]. This effect enables consumers to make more rational purchase decisions, which leads to transactions and increased consumption. Meanwhile, digital industrialization has brought about the rapid development of online shopping platforms and digital logistics supply chains. This can effectively reduce the geographical boundaries and barriers of goods and services. Consumers' shopping needs can be met faster and safer, and the time per purchase is shortened [44].

For another, through deep integration with traditional industries, the digital economy realizes digital transformation and accelerates the digital transformation of the consumption supply side [45]. Relying on the new division of labor mode and production mode formed by digital technology, market entities have significantly improved production efficiency and value chain, and ultimately increased the effective supply of the whole society [46]. The consumer demand can be better matched with consumption supply. Consumers' constantly upgrading consumption demand can be quickly and effectively satisfied, which is conducive to achieving high-quality development [47]. Consequently, hypotheses 3 and 4 were proposed:

H3: Under the regulation of consumption, there is a threshold effect on the impact of the digital economy on high-quality economic development.

H4: Consumption expansion promotes high-quality economic development through the digital economy.

## Methodology

### Models

Firstly, we constructed the benchmark individual fixed-effect model:

$$hqe_{it} = \alpha_0 + \alpha_1 digit_{it} + \alpha_2 digit_{it}^2 + \alpha_3 control_{it} + u_i + \varepsilon_{it}, \qquad (1)$$

where $hqe_{it}$ indicates the level of regional high-quality development index of the region $i$ in the period of $t$. $\alpha_0$ is the constant. $digit_{it}$ is the digital economy index and $digit_{it}^2$ is the square of $digit_{it}$. $control_{it}$ represents the control variables. $u_i$ indicates individual fixed effects and $\varepsilon_{it}$ is the random error.

The mediational effect model of digit economy was defined as follows.

$$hqe_{it} = \beta_0 + \beta_1 cd_{it} + \beta_2 control_{it} + u_i + \varepsilon_{it} \qquad (2)$$

$$digit_{it} = \gamma_0 + \gamma_1 cd_{it} + \gamma_2 control_{it} + u_i + \varepsilon_{it} \qquad (3)$$

$$hqe_{it} = \delta_0 + \delta_1 cd_{it} + \delta_2 digit_{it} + \delta_3 control_{it} + u_i + \varepsilon_{it} \qquad (4)$$

Furthermore, we conducted the static panel threshold regression method. The threshold regression model is designed for non-dynamic panels with individual fixed effects, and refers to an econometric research method that causes structural mutations in another economic parameter when it reaches a certain value [48]. This method determines the threshold value by minimizing the sum of squares of the residuals. It uses a fixed-effect transformation and obtains the regression slope through least squares estimation. Taking use of the non-standard asymptotic theory for inference, this method allows to construct the confidence intervals for hypothesis testing, thereby overcoming the bias caused by the mutation of the subjective setting architecture.

Considering the theoretical analysis above, the following panel threshold model was constructed:

$$hqe_{it} = \theta_0 + \theta_1 digit_{it} \cdot I(cd_{it} \leq \mu_1) + \theta_2 digit_{it} \cdot I(\mu_1 \leq cd_{it} \leq \mu_2) + \theta_3 digit_{it} \cdot I(cd_{it} \geq \mu_2) + \theta_4 control_{it} + u_i + \varepsilon_{it} \qquad (5)$$

$$hqe_{it} = \varphi_0 + \varphi_1 digit_{it} \cdot I(cs_{it} \leq \sigma_1) + \varphi_2 digit_{it} \cdot I(\sigma_1 \leq cs_{it} \leq \sigma_2) + \varphi_3 digit_{it} \cdot I(cs_{it} \geq \sigma_2) + \varphi_4 control_{it} + u_i + \varepsilon_{it} \qquad (6)$$

Eqs (5) and (6) indicate that there is a double threshold effect on the impact of the digital economy on high-quality economic development. $cd_{it}$ and $cs_{it}$ indicate the threshold variable consumption demand and consumption supply respectively. $digit_{it}$ means the district dependent variable, which is the core explanatory variables affected by the threshold variable. $\mu_i$ and $\sigma_i$ is the threshold to be estimated. $I(\cdot)$ is the indicative function, and $control_{it}$ represents the control variables.

Explained variable: high quality economic development index ($hqe$). The measurement of high-quality economic development is diversified and comprehensive. We constructed the index system from the five perspectives of innovation, coordination, green, openness and sharing [49,50]. The entropy weight method was used to measure the index weight of the high-quality economic development index, and the results are shown in Table 1.

Explanatory variable: Regional Digital Economy Index ($digit$). We made use of the index system to measure the digital economy index, which was constructed from four dimensions: digital foundation, digital industry, digital integration, and digital innovation [51–54]. The

**Table 1. The index system of the high-quality economic development index.**

| Dimensions | Primary Indicators | Secondary Indicators | Symbol |
|---|---|---|---|
| Innovation (0.2409) [a] | Innovation Input (0.0829) | R&D Input | + |
|  | Innovation Output (0.1579) | Number of Patents Granted | + |
| Coordination (0.1016) | Urban-rural Coordination (0.0432) | Proportion of Urban Population | + |
|  | Industry Coordination (0.0584) | Industry Advanced Coefficient | + |
| Green (0.1039) | Energy Consumption (-0.0256) | Total Electricity Consumption | − |
|  | Environmental Governance (0.1295) | Comprehensive Utilization of Industrial Solid Waste | + |
| Openness (0.3027) | Foreign Trade (0.2015) | Total Export-Import Volume | + |
|  | Foreign Direct Investment (FDI) (0.1012) | Foreign Direct Investment | + |
| Sharing (0.1996) | Economic Sharing (0.1353) | Total Retail Sales of Consumer Goods | + |
|  | Social Sharing (0.0643) | Number of Beds in Health Institutions | + |

[a] The indicator weight calculated by the entropy weight method.

index weight was calculated by the entropy weight method, and the results are shown in Table 2.

Control variables: In order to avoid the problem of multicollinearity due to too many control variables, we referred to previous researches and considered factors that may affect the high-quality economic development in the model as follows. (1) Degree of openness to the foreign countries (*open*), measured by the share of FDI in Gross Domestic Product (GDP). (2) Infrastructure construction (*basein*), measured by public transport vehicles per 10,000 people

**Table 2. The index system of the regional digital economy.**

| Dimensions | Primary Indicators | Secondary Indicators | Symbol |
|---|---|---|---|
| The Digital Economy | Digital Foundation (0.2166) [a] | Number of Mobile Phone Base Stations | + |
|  |  | Internet Broadband Access Port | + |
|  |  | Mobile Phone Penetration | + |
|  | Digital Industry (0.2815) | Investment in Fixed Assets of Electronic Information Manufacturing | + |
|  |  | Software Product Revenue Scale | + |
|  |  | Information Technology Services Revenue | + |
|  |  | Volume of Telecommunication Service | + |
|  | Digital Integration (0.2888) | Proportion of Enterprises Engaged in E-commerce Transactions | + |
|  |  | E-commerce Sales | + |
|  |  | Digital Financial Inclusion Index | + |
|  |  | Online Government Service Capability Index | + |
|  | Digital Innovation (0.2131) | Number of Patent Applications Granted | + |
|  |  | Number of High School Students | + |
|  |  | Number of R&D Projects of Industrial Enterprises Above Designated Size | + |

[a] The indicator weight calculated by the entropy weight method.

**Table 3. Descriptive statistics of variables.**

| Variable | Mean | Standard Deviation | Min | Max | Observation |
|---|---|---|---|---|---|
| hqe | 0.376 | 0.093 | 0.234 | 0.795 | 372 |
| digit | 0.332 | 0.129 | 0.114 | 0.934 | 372 |
| cd | 0.379 | 0.066 | 0.222 | 0.603 | 372 |
| cs | 0.109 | 0.112 | 0.005 | 0.715 | 372 |
| open | 0.388 | 0.402 | 0.050 | 3.290 | 372 |
| basein | 12.410 | 3.119 | 6.200 | 26.550 | 372 |
| govern | 0.281 | 0.208 | 0.110 | 1.380 | 372 |

(standard station). (3) Government functions (*govern*), measured by the proportion of fiscal expenditure in total GDP.

Threshold variables: For consumption expansion, we analyzed the threshold effect of the digital economy in the path of high-quality economic development from the two aspects of consumer demand and consumption supply. The proportion of total household consumption to GDP was introduced as a proxy indicator of consumption demand (*cd*). As for the consumption supply (*cs*), we took the share of e-commerce turnover in GDP as the proxy variable.

## Data

We selected data of 31 provinces in China from 2010 to 2021 (excluding Hong Kong, Macao and Taiwan). Data sources include China Statistical Yearbook, China Information Industry Yearbook, China Industrial Statistical Yearbook and the statistical yearbooks and statistical bulletins of the provinces. Missing data was interpolated by the expected mean. In order to eliminate the influence of dimensions and alleviate the estimation bias caused by the problem of heteroscedasticity, all variable indicators were made logarithmic to further enhance the stationarity of the data. The results of the descriptive statistics for variables are shown in Table 3.

## Empirical results

### Regression analysis and robustness testing

Firstly, based on the results of the Hausman test [55], we determined to implement the fixed-effect model. Among the individual fixed-effect models, the time fixed-effect model, and the individual-time dual fixed-effect model, the result was found to be most significant by using the individual fixed-effect model. The results of regression analysis are shown in the model (1) in Table 4.

Further, the robustness test was carried out by shrinking the sample data, lagging the core explanatory variable by one period, and introducing the interactive terms of province and year. The results are shown in Table 4: Model (2) through Model (4).

In model (1), we can find that the digital economy and the high-quality development show a positive correlation at the significance level of 1%. The rapid development of the digital economy effectively promotes high-quality economic development. This verifies hypothesis 1. Meanwhile the coefficient of $digit^2$ are always negative. The relationship between the digit economy and the high-quality economy is inverted U-shape, which verifies hypothesis 2. In the robustness test, the regression coefficients of the digital economy in models (2), (3) and (4) all positively affect the high-quality economic development at the significance level of 1%, which verifies the robustness of the regression results of the individual fixed-effect model.

The control variables *open* are all significant at the 5% level, indicating that the international trade and investment have an impact on the high-quality economic development. In contrast,

**Table 4. Individual fixed-effect model results and robustness tests.**

| Variables | Model (1) | Model (2) | Model (3) | Model (4) |
|---|---|---|---|---|
| *digit* | 0.8051***[a] (0.0313) [b] | 0.7267*** (0.0296) | | 0.7892*** (0.0459) |
| *L.digit* | | | 0.7062*** (0.0368) | |
| *digit²* | -0.1690*** (0.0354) | -0.1783*** (0.0335) | | -0.1572*** (0.0435) |
| *L.digit²* | | | -0.1394*** (0.0454) | |
| *open* | 0.0026*** (0.0003) | 0.0025*** (0.0003) | 0.0024*** (0.0004) | 0.0026** (0.0003) |
| *basein* | 0.0158 (0.0216) | 0.0311 (0.0185) | 0.0136** (0.0200) | 0.0296 (0.0062) |
| *govern* | -0.0302*** (0.0243) | -0.0203*** (0.0071) | -0.0221** (0.0081) | -0.0295*** (0.0084) |
| *province×year* | | | | 0.0102*** (0.0216) |
| *constant* | 0.2083*** (0.0243) | 0.2037*** (0.0234) | 0.2505*** (0.0274) | -1.5805*** (0.5775) |
| *Obs* | 372 | 310 | 341 | 372 |
| *R²* | 0.9018 | 0.8889 | 0.8871 | 0.9223 |

[a] *, **, *** indicate a significance level of 10%, 5% and 1%, respectively.

[b] Values in () are standard errors.

the regression coefficient of *govern* is -0.0302, indicating that government functions will have inhibitory effect on high-quality economic development. In the process of high-quality economic development, the increase in the proportion of fiscal expenditure will have an obvious crowding out effect. The degree of base installation has a positive effect on high-quality economic development, but the coefficient is not significant.

To avoid the multicollinearity problem, the Variance Inflation Factor (VIF) expansion factor test [56] was performed for all explanatory variables. Empirically, there is no serious collinearity issue when the maximum VIF value is less than 10. The result is shown in Table 5. It is proved that there is no multicollinearity problem.

## Mediation effect analysis

The digit economy and the high-quality development are verified as the inverted U-shaped relationship. When considering expanding consumption, the core factors and paths of influence for high-quality economic development may undergo new changes. We examined the intermediary effect of consumer expansion on the impact of digital economy on high-quality

**Table 5. Multicollinearity test.**

| Variables | VIF | 1/VIF |
|---|---|---|
| *digit* | 1.40 | 0.714 |
| *open* | 1.01 | 0.776 |
| *basein* | 1.14 | 0.878 |
| *govern* | 1.29 | 0.994 |
| *MEAN* | 1.21 | |

**Table 6. The intermediary model results of consumption expansion.**

| Variables | hqe | cd | hqe |
|---|---|---|---|
| digit | 0.6652*** [a] <br> (0.0114) [b] | 0.3627** <br> (0.0316) | 0.5578*** <br> (0.0118) |
| cd | | | 0.4453** <br> (0.0188) |
| open | 0.0027*** <br> (0.0003) | 0.0002 <br> (0.0008) | 0.0027*** <br> (0.0086) |
| basein | 0.0059 <br> (0.0062) | 0.0924*** <br> (0.0171) | 0.0017 <br> (0.0064) |
| govern | -0.0256*** <br> (0.0084) | -0.0619*** <br> (0.0233) | -0.0228*** <br> (0.0085) |
| constant | 0.2002*** <br> (0.0250) | 0.3039*** <br> (0.0689) | 0.1864*** <br> (0.0256) |
| $R^2$ | 0.8998 | 0.1567 | 0.9214 |

[a] *, **, *** indicate a significance level of 10%, 5% and 1%, respectively.

[b] Values in () are standard errors.

development. By separately analyzing consumer demand and consumption supply as mediating factors, it was found that only the mediation effect of consumer demand is significant. The results are shown in Table 6.

It is shown that the coefficients of the explanatory variables are significant at the 5% level, indicating the existence of the mediating effect of consumption demand. Among the influence paths of the digital economy on the high-quality development, the partial intermediary effect of consumption demand exists, accounting for 16.15% of the total effect. Then, we analyzed the impact of consumption expansion on the high-quality development, considering the digit economy as the mediator. The results are shown in Table 7.

When the digital economy was used as an intermediary variable, there is a complete intermediary effect in the consumption supply to promote high-quality development. And there is

**Table 7. The intermediary model results of digital economy.**

| Variables | (1) | (2) | (3) | (4) | (5) | (6) |
|---|---|---|---|---|---|---|
| | hqe | digit | hqe | hqe | digit | hqe |
| cd | 0.3411*** [a] <br> (0.0604) [b] | 0.4492*** <br> (0.1315) | 0.0456** <br> (0.0196) | | | |
| cs | | | | 0.4558*** <br> (0.0375) | 0.7039*** <br> (0.0518) | -0.0193 <br> (0.0168) |
| digit | | | 0.6578*** <br> (0.0118) | | | 0.6749*** <br> (0.0142) |
| open | 0.0041*** <br> (0.0009) | 0.0020* <br> (0.0011) | 0.0027*** <br> (0.0003) | 0.0029*** <br> (0.0008) | 0.0003 <br> (0.0012) | 0.0028*** <br> (0.0003) |
| basein | -0.0002 <br> (0.0205) | -0.0029*** <br> (0.0421) | 0.0017 <br> (0.0064) | 0.0485*** <br> (0.0171) | 0.0648*** <br> (0.0237) | 0.0048 <br> (0.0063) |
| govern | 0.1154*** <br> (0.0259) | 0.2101*** <br> (0.0686) | -0.0228** <br> (0.0085) | 0.0272 <br> (0.0235) | 0.0762** <br> (0.0324) | -0.0242*** <br> (0.0086) |
| constant | -0.1874** <br> (0.0787) | -0.5683*** <br> (0.1698) | 0.1864*** <br> (0.0256) | 0.0499 <br> (0.0688) | -0.2193** <br> (0.0952) | 0.1981*** <br> (0.0251) |
| $R^2$ | 0.4109 | 0.3854 | 0.8975 | 0.3885 | 0.4266 | 0.9043 |

[a] *, **, *** indicate a significance level of 10%, 5% and 1%, respectively.

[b] Values in () are standard errors.

**Table 8.  Results of threshold effect test.**

| Explanatory Variables | Threshold Variables | Number of Threshold | *F* Value | *P* Value | Number of *BS* | Critical Value | | |
|---|---|---|---|---|---|---|---|---|
| | | | | | | 1% | 5% | 10% |
| The Digital Economy | Consumer Demand | Single | 9.31*** [a] | 0.000 | 300 | 6.582 | 4.757 | 4.048 |
| | | Double | 7.80*** | 0.023 | 300 | 9.652 | 6.749 | 5.642 |
| | | Triple | 5.35 | 0.673 | 300 | 23.804 | 17.324 | 10.789 |
| | Consumption Supply | Single | 15.45*** | 0.000 | 300 | 11.384 | 9.295 | 8.553 |
| | | Double | 6.59 | 0.383 | 300 | 12.247 | 10.854 | 9.573 |
| | | Triple | 4.73 | 0.451 | 300 | 22.559 | 12.074 | 7.221 |

[a] *\*,\*\*,\*\*\* indicate a significance level of 10%, 5% and 1%, respectively.*

a partial intermediary effect in the path of consumer demand influence. Hypothesis 4 is proved. In the model (1) and (4), both consumer demand and consumption supply can improve the quality of economic development. Model (2) and (5) demonstrate that digit economy promotes the consumption expansion. The coefficients of the mediating variables are significant at 1% significance level in model (3) and (6). Model (3) shows that the intermediary effect of the digital economy accounts for 86.63% of the total effect. Most of the promotion effect is realized through the digital economy. But the coefficient of the consumption supply in model (6) is insignificant. It is explained that the consumption supply completely relies on the digit economy to affect the high-quality development.

## Threshold effect test and model results

In order to further explore the nonlinear influence of digital economy on China's high-quality economic development, we introduced consumption expansion to analyze the threshold effect. The bootstrap number was selected as 300. The proportion of outliers removed within the threshold group was 0.05, and the number of grids calculated by the sample network was 300. The threshold effect test was carried out in combination with clustering robust standard error. The threshold effect test results are shown in Table 8, and the threshold estimates and confidence interval results are shown in Table 9.

The results show that when the consumer demand is tested as a threshold variable, the impact of the digital economy on high-quality economic development passes the test of 1% significance level, indicating that there is a double threshold effect. When consumption supply is tested as a threshold variable, the impact of the digital economy on high-quality economic development is also significant at the level of 1%, but there is only a single threshold effect. Hypothesis 3 is validated.

Table 9 shows the thresholds and their confidence intervals. For the high-quality economic development, when the digital economy is considered as the core explanatory variable, the first threshold value of consumer demand is -1.0244, and the second threshold value is -1.0016. The first threshold for consumption supply is -3.6889, and the second threshold is not reported.

**Table 9.  Threshold estimates for each threshold variable.**

| Explanatory Variables | Threshold Variables | Number of Thresholds | Estimated Value | Confidence Interval |
|---|---|---|---|---|
| The Digital Economy | Consumer Demand | Single | -1.0244 | [-1.1109, -1.0217] |
| | | Double | -1.0016 | [-1.0016, -0.9989] |
| | Consumption Supply | Single | -3.6889 | [-4.0178, -3.5543] |

**Table 10. Results of sample threshold regression.**

| Variables | Threshold Variables | |
|:---:|:---:|:---:|
| | *cd* | *cs* |
| *open* | 0.0053 (0.97) | 0.019*** (3.947) |
| *basein* | 0.0.136 (0.63) | 0.0084 (0.41) |
| *govern* | -0.036*** [a] (-4.45) [b] | -0.0489*** (-4.21) |
| *digit_0* | 0.3974*** (29.61) | 0.3381*** (20.91) |
| *digit_1* | 0.7415*** (24.52) | 0.6758*** (24.49) |
| *digit_2* | 0.4484*** (19.83) | |
| *constant* | -0.0377 (-0.43) | -0.1312 (-1.30) |

[a] *, **, *** indicate a significance level of 10%, 5% and 1%, respectively.

[b] Values in () are standard errors.

After determining the threshold value, panel threshold regression was performed on the sample data, and the results are shown in Table 10. When consumer demand is below the first threshold of -1.0244, the coefficient of the digital economy is 0.3974. When consumer demand is between -1.0244 and -1.0016, the impact coefficient of the digital economy on high-quality economic development increased to 0.7415. This impact coefficient drops to 0.4484 when consumer demand is above the threshold of -1.0016. All coefficients passed the 1% significance level test. This result validates hypotheses 2 and 3.

This shows that when the proportion of total household consumption in GDP is low and consumer demand is insufficient, the increase in consumer demand will significantly improve the digital economy in promoting high-quality economic development. This is because the growing consumer demand is conducive to the basic support for digital industrialization and industrial digitalization in the digital economy to improve the quality of economic growth. In giving full play to the role of the digital economy in high-quality economic development, consumer demand is conducive to releasing the potential of the digital economy and strengthening the enabling role of the digital economy in high-quality economic development. However, it should be noted that when consumer demand increases, the proportion of total household consumption in GDP is high and consumer demand is close to saturation, and further increasing consumer demand will inhibit the role of the digital economy in promoting high-quality economic development.

With the increase in consumer demand, the utility of the digital economy to empower high-quality economic development presents an inverted U-shaped structure. In the benchmark model, we verified the inverted U-shaped structure between the digital economy and the high-quality development. Analyzed from the perspective of consumption expansion, the inverted U-shaped structure may be affected by consumer demand. The validation result of hypothesis 2 is strengthened.

Therefore, when utilizing consumption to stimulate the role of the digital economy in high-quality economic development, we should grasp the principle of moderation. Only by controlling consumption within the suitable range, the ideal effect of promoting high-quality development can be achieved.

When consumption supply is analyzed as a threshold variable, it was found that when the consumption supply is lower than the threshold value of -3.6889, the regression coefficient of the digital economy is 0.3381 and passes the significance test at the significance level of 1%. When the consumption supply exceeds the threshold value of -3.6889, the coefficient of the digital economy increases to 0.6758, which is significant at the level of 1%.

This suggests that the role of the digital economy in promoting high-quality economic development has an increasing marginal utility with the support of consumption supply, and there is a coordinated relationship among the three. The rapid development of the digital economy has provided a good digital industry foundation for consumption supply. With the deepening of the digitalization of the supply industry, this effect on the empowerment of high-quality economic development will strengthen continuously. The consumption supply capacity improves, and the development potential of the digital economy continues to be released, which will bring about sustained growth of high-quality economic development.

## Heterogeneity analysis

There is an imbalance in the economic development of different regions in China. In addition, the level of development of the digital economy varies widely across regions. To further explore the regional differential impact of the digital economy on high-quality economic development, and the threshold effect of consumption expansion on high-quality economic development empowered by the digital economy among regions, we divided 31 provinces into three regions according to the classification method of China's National Bureau of Statistics, and estimated the fixed effect regression respectively. The eastern region includes 11 regions: Beijing, Tianjin, Hebei, Liaoning, Shanghai, Jiangsu, Zhejiang, Fujian, Shandong, Guangdong, and Hainan; the central region includes 8 regions: Shanxi, Jilin, Heilongjiang, Anhui, Jiangxi, Henan, Hubei, and Hunan; the western region includes 12 regions: Inner Mongolia, Guangxi, Chongqing, Sichuan, Guizhou, Yunnan, Tibet, Shaanxi, Gansu, Qinghai, Ningxia, and Xinjiang. At the same time, the threshold effect of consumption expansion in each region was examined, and the results are shown in Table 11.

The results show that the development level of the digital economy in the eastern, central, and western regions has a significant positive effect on high-quality economic development. Among them, the most obvious impact effect is in the central region, followed by the eastern region, and the weakest role of the development level of the digital economy on the high-quality economic development is in the western region, which reflects the existence of the digital divide.

This result differs from the literature [57]. This may be caused by the samples. The literature analyzed based on 30 cities of China. There is randomness in the sample distribution. We included all cities in the sample on a provincial basis, which combine the effects of urban heterogeneity. The economic development level in the eastern region is much higher than other regions, and the level of digital economy development is also relatively high. Therefore, the empowering potential of the digital economy for high-quality economic development has been more effectively released. In contrast, the development of the digital economy in the central region is relatively short, starting later than in the eastern region. At present, the central region is in a period of rapid development of the digital economy, and the development potential of the digital economy is great, which is most effective in enabling the high-quality development of the regional economy. The economy in the western region is relatively backward, and the overall level of digital economy development is limited, but the potential for digitalization is huge. Given the current level of digitalization, although the western region has a significant positive impact on the high-quality development of the economy, the effect is relatively limited.

**Table 11. Results of regional heterogeneity regression.**

| Models | | Core Explanatory Variables | | | | Control Variables | Obs |
|---|---|---|---|---|---|---|---|
| | | *digit* | *digit_0* | *digit_1* | *digit_2* | | |
| East | *FE* | 0.4094*** [a]<br>(0.0673) [b] | | | | YES | 132 |
| | *cd* | | 0.3593***<br>(0.0663) | 0.4008***<br>(0.0726) | 0.6677***<br>(0.1161) | YES | 132 |
| | *cs* | | 0.3795***<br>(0.0638) | 0.6171***<br>(0.0736) | | YES | 132 |
| Central | *FE* | 0.5407***<br>(0.0667) | | | | YES | 96 |
| | *cd* | | 0.5376*<br>(0.1147) | 0.3211*<br>(0.1085) | 0.1992<br>(0.1048) | YES | 96 |
| | *cs* | | 0.2096**<br>(0.0752) | 0.4501**<br>(0.0747) | 0.6914***<br>(0.0792) | YES | 96 |
| West | *FE* | 0.1887***<br>(0.0811) | | | | YES | 144 |
| | *cd* | | 0.1663***<br>(0.0817) | 0.4060***<br>(0.0885) | 0.2650***<br>(0.0864) | YES | 144 |
| | *cs* | | 0.2671***<br>(0.0285) | 0.3985***<br>(0.0318) | | YES | 144 |

Note: *,**, and *** indicate a significance level of 10%, 5%, and 1%, respectively. Values in () are t values.

In addition, when examining the threshold regulating role of consumption expansion, it was found that there are two threshold values for consumer demand in the eastern region. When consumer demand is lower than the first threshold, the impact coefficient of the digital economy on high-quality development is 0.3593. When consumer demand is between the first and second thresholds, the impact coefficient of the digital economy on high-quality development increases to 0.4008, and as consumer demand crosses the second threshold, the impact coefficient increases to 0.6677. There is one threshold value for consumption supply. With the continuous increase of consumption supply, the enabling effect of the digital economy on high-quality development has increased from 0.3795 to 0.6171. The above results indicate that in the eastern region of China, with the continuous improvement of consumer demand and consumption supply, the enabling role of the digital economy in high-quality economic development has been continuously enhanced, and the impact effect has shown a marginal increasing development trend. This is due to the long-term strong and rapid economic development of the eastern region, while the demand structure and industrial supply structure have been continuously optimized and adjusted. In the eastern region, there is still much room for the potential development of the digital economy. Based on the advantages of a large number of resource reserves and policy support, the eastern region will make use of the digital economy to drive the common development of other regions.

In the central region, consumer demand and consumption supply both exist two thresholds. Among them, when consumer demand is lower than the first threshold, the impact coefficient of the digital economy on high-quality development is significant at the 10% level, which is 0.5376. While consumer demand is between the first and second thresholds, the impact coefficient drops to 0.3211. This impact coefficient further drops to 0.1992 when consumer demand is higher than the second threshold, but it is not significant even though the significance level is 0.1. With the continuous improvement of consumption supply and crossing the two thresholds, the impact coefficients of the digital economy on high-quality economic development increase. The above threshold model regression results prove that with

the increase of consumer demand in the central region, the enabling effect of the digital economy on high-quality economic development shows a marginal diminishing effect.

This suggests that the expansion of consumer demand in the central region has not effectively stimulated the release of the potential of the digital economy. Instead, it has led to a weakening of the digital economy's support for high-quality economic development, while consumer demand has increased. The reason may lies in that although the pace of digital economy development in central region has accelerated rapidly in recent years, in the process of developing the digital economy, the contradiction between industrial and supply-demand structures has gradually become prominent. The incongruity of industrial structure and the low matching between supply and demand lead to constraints on the development of industrial digitalization and digital industrialization. The growth of consumer demand stimulated by the digital economy cannot be satisfied quickly and efficiently in the short term, thus it is shown as the marginal reduction in the enabling effect. From the perspective of consumption supply, the increase in consumption supply provides a more sufficient supply of resources for the digital economy, which will make the digital economy continuously strengthen its enabling effect in high-quality economic development, showing obvious marginal incremental effects.

Last but not least, there are two thresholds for consumer demand in the western region. The impact coefficient of the digital economy on high-quality economic development is 0.1663 when consumer demand is at a low level. When consumer demand is between the first and second thresholds, the impact coefficient rises to 0.4060. However, when consumer demand is above the second threshold, the impact coefficient drops to 0.265. All coefficients are significant at the 1% level. It is introduced that in the western region, with the increase of consumer demand, the enabling effect of the digital economy on high-quality development shows an inverted "U" shaped effect of increasing marginally and then decreasing marginally. When the level of consumer demand is low, the consumption potential has not yet been fully released, and the digital economy will promote high-quality economic development and play an obvious enabling role. However, when the level of consumer demand is higher than the second threshold, the enabling role seems to be insufficient. This is mainly due to the relatively limited level of the digital economy in the western region, and its positive effect on high-quality economic development is easy to work in the initial stage of digital economy development. However, the further development of the digital economy is constrained by the limitations of industrial foundation, industrial structure, and resource level. As a result, after the digital economy stimulates the release of consumer demand, it cannot meet the new demand in the short term, and its empowerment capacity for high-quality economic development has declined marginally. There is only one threshold for the threshold effect of consumption supply. With the increase in consumption supply, the impact of the digital economy on high-quality development strengthens.

In general, in the eastern, central, and western regions, the threshold effect of consumption supply is positive for the digital economy and high-quality growth while the threshold effect of consumption demand has significant heterogeneity characteristics.

## Conclusions

Based on the entropy weight method to construct the digital economy development index and the economic high-quality development index, this paper studied the impact of the digital economy and consumption expansion on high-quality development. The non-linear relationship, the mediator effect and the threshold effect were explored in the empirical analysis respectively. Given the existence of the digital divide, the heterogeneity of threshold effects was further discussed. The results are shown as follows:

1. The development of the digital economy can have a significant positive effect on the high-quality development. But this promotion has an inflection point, which is manifested as an inverted U-shaped structure.

2. In the mediator effect analysis, the consumer demand has a partial mediating effect while the consumption supply has not. Most of the impact of consumer demand on high-quality development needs to be realized through the intermediary effect of the digital economy. Consumer supply is achieved entirely through the digital economy to drive high quality development.

3. In the expansion of consumption, consumer demand and consumption supply have a threshold effect on the digital economy to empower the high-quality development. The impact varies in the two threshold variables. The threshold effect of consumer demand leads to the inverted "U" shape while the effect of consumption supply on the digital economy and the high-quality development of the economy is always positive.

4. Considering regional heterogeneity, the digital economy in the central region has the most obvious effect on promoting high-quality economic development, while the weakest effect exists in the western region. When considering the threshold effect of consumption expansion, the threshold effect of consumption supply in the eastern, central and western regions is the same, which is manifested in the marginal increasing effect on the high-quality development of the digital economy empowerment economy within different threshold ranges. However, in different regions, the threshold effect of consumer demand is completely different. The growth of consumer demand in the eastern region will continuously strengthen the role of the digital economy in empowering high-quality economic development, and the two will develop in tandem. In the central region, the growth of consumer demand will inhibit the enabling effect of the digital economy and show marginal incremental effects. The western region is reflected in the inverted "U" shaped empowerment effect.

Based on the above analysis, in order to give full play to the potential and value of the digital economy and achieve sustainable and high-quality economic development in other countries, we proposed the following suggestions:

1. In order to achieve sustainable and high-quality economic development, accelerating the industrial digitalization and digital industrialization of the digital economy is necessary. The government needs to create a good policy environment for the development of the digital economy, and promote the construction of digital industry infrastructure.

2. Fully recognize the role of consumption supply capacity in high-quality development. While developing the digital economy and promoting high-quality economic development, we should pay attention to the improvement of consumption supply capacity and the matching degree between supply capacity and consumer demand.

3. Advantageous regions should be allowed to give full play to their exemplary role and release spillover effects to other regions, so as to drive the overall development of the domestic digital economy and high-quality economic development.

4. When the other countries develop the digital economy and promote sustained and high-quality economic development, they should pay more attention to industrial restructuring and industrial structure upgrading. It is suggested to focus on the shift from the original factor-driven and investment-driven extensive development mode to the sustainable development model of scientific and technological innovation. Only by unleashing the potential of

the digital economy can the sustainable and high-quality economic development be promoted.

Finally, this study still has limitations and future directions. The spatial econometric models are not utilized to analyze the spatial impact of digital economy and consumption expansion on high-quality development. In addition, how to measure consumption expansion more comprehensively is also worth exploring. Consumption expansion is only one aspect of consumption growth, the consumption quality and its impact on digital economy and high-quality development also deserves further study.

## Supporting information

**S1 Dataset.**
(XLSX)

## Author Contributions

**Conceptualization:** Xiaoxuan Li.

**Data curation:** Qi Wu.

**Funding acquisition:** Xiaoxuan Li.

**Methodology:** Xiaoxuan Li, Qi Wu.

**Software:** Qi Wu.

**Writing – original draft:** Xiaoxuan Li.

**Writing – review & editing:** Qi Wu.

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
