## [Decision Letter · Decision Letter 0]

29 Aug 2023

PONE-D-23-22047The Impact of Digital Economy on High-quality Economic Development: Research Based on the Consumption ExpansionPLOS ONE

Dear Dr. Li,

Thank you for submitting your manuscript to PLOS ONE. After careful consideration, we feel that it has merit but does not fully meet PLOS ONE’s publication criteria as it currently stands. Therefore, we invite you to submit a revised version of the manuscript that addresses the points raised during the review process.

We look forward to receiving your revised manuscript.

Kind regards,

Jing Cheng

Academic Editor

PLOS ONE

Journal Requirements:

"This research was funded by the Youth Project of Anhui Natural Science Foundation (No. 1908085QG305), the key project of humanities and social science research in colleges and universities in Anhui Province (No. SK2020A0341)."

5. We note that all Figures in your submission contain [map/satellite] images which may be copyrighted. All PLOS content is published under the Creative Commons Attribution License (CC BY 4.0), which means that the manuscript, images, and Supporting Information files will be freely available online, and any third party is permitted to access, download, copy, distribute, and use these materials in any way, even commercially, with proper attribution. For these reasons, we cannot publish previously copyrighted maps or satellite images created using proprietary data, such as Google software (Google Maps, Street View, and Earth). For more information, see our copyright guidelines: http://journals.plos.org/plosone/s/licenses-and-copyright.

a. You may seek permission from the original copyright holder of all Figures to publish the content specifically under the CC BY 4.0 license.  

6. Please ensure that you refer to all Figures in your text as, if accepted, production will need this reference to link the reader to the figure.

7. Please include a separate caption for each figure in your manuscript.

Reviewers' comments:

Reviewer's Responses to Questions

**Comments to the Author**

1. Is the manuscript technically sound, and do the data support the conclusions?

Reviewer #1: No

Reviewer #2: Yes

2. Has the statistical analysis been performed appropriately and rigorously? 

Reviewer #1: No

Reviewer #2: Yes

3. Have the authors made all data underlying the findings in their manuscript fully available?

Reviewer #1: No

Reviewer #2: Yes

4. Is the manuscript presented in an intelligible fashion and written in standard English?

Reviewer #1: No

Reviewer #2: No

5. Review Comments to the Author

Reviewer #1: Comments on Manuscript “The Impact of Digital Economy on High-quality Economic Development: Research Based on the Consumption Expansion” Thank you very much for giving me the opportunity to review the manuscript.

Studies have found that the digital economy plays a positive role in promoting high-quality economic development. The contemporary economy pursuits the high-quality sustainable development of innovation, coordination, green, openness and sharing. The digital dividends brought by digital economy can well help to promote high-quality economic development. Meanwhile, the digital industrialization and industrial digitalization have spawned new consumer demand and consumption supply modes. It is necessary to analyze the role of consumption expansion in the impact of digital economy on high-quality economic development. Based on Chinese provincial panel data, we first apply the entropy weight method to construct digital economy index and high-quality economic development index. On this basis, it is verified that the development of the digital economy can positively promote the high-quality development of the economy. Then we use the threshold effect model to analyze the role of consumer demand and consumption supply in the digital economy empowering the high-quality development. Regional heterogeneity of this effect is further taken into account. The results dedicate that the digital economy can effectively promote high-quality economic development. This also can be affected by the threshold of consumption expansion, which is manifested in the marginal incremental effect due to the growth of consumption supply. On the contrary, the growth of consumer demand has led to the inverted U-shape of the digital economy to promote high-quality economic development. In the heterogeneity analysis, the threshold effect also varies greatly. The research enriches the theoretical achievements and reveals the impact of consumption expansion on the digital economy affecting the high-quality development, which has certain references for other countries and regions. I have reviewed this paper thoroughly and a few suggestions are given below:

Literature Review:

This section is well-aligned and presents an effective blend of recent and past studies. Overall, this section is appropriate in my opinion. Kindly add some studies in literature review section;

https://doi.org/10.1007/s11356-022-19718-6

https://doi.org/10.3390/su14031054

https://doi.org/10.1007/s11356-021-17438-x

https://doi.org/10.1002/ijfe.2073

https://doi.org/10.1108/FS-02-2021-0053

https://doi.org/10.1007/s11356-022-19954-w

https://doi.org/10.1007/s11356-022-20922-7

https://doi.org/10.1007/s11356-022-20178-1

https://doi.org/10.3389/fenvs.2022.967418

https://doi.org/10.3389/fpsyg.2022.892488

https://doi.org/10.1007/s11356-023-27473-5

https://doi.org/10.1108/LHT-03-2021-0113

https://doi.org/10.1016/j.resourpol.2022.102730

https://doi.org/10.1007/s11356-022-19628-7

https://doi.org/10.1007/s11356-022-21929-w

https://doi.org/10.1177/21582440211061554

Study design:

This section is well-written and well-explained. Referring to my comments in the Introduction section, try to explain each abbreviation at first and then use its short form.

Empirical Results:

Results are explained in detail. Must add much more explanations and interpretations for the results, which are not enough. It is suggested to compare the results of the present research with some similar studies which is done before (more justification is needed).

Future Directions:

Please revise your future research and limitation part into more detail. It would be best if you enhanced your this section.

I hope these comments would enhance the quality of the manuscript to make it an appropriate fit for the Journal readership.

Reviewer #2: Please include the following items when submitting your revised manuscript

1. In order to establish the significance and value of the study, it is necessary to provide a comprehensive rationale for the research, which emphasizes its relevance and unique contributions to the current scholarly discourse. This will strengthen the study's originality and scholarly impact.

2. English check/editing: I found minor typos/ grammatical, this study was presented with depraved English language usage.

3. The conclusion is sufficiently discussed with some useful policy implications. However, limitations and future directions of research should have a reasonably sized paragraph.

6. PLOS authors have the option to publish the peer review history of their article (what does this mean?). If published, this will include your full peer review and any attached files.

Reviewer #1: **Yes: **KASHIF ABBASS

Reviewer #2: **Yes: **Dr. SHABBIR AHMED Department of Economics, Govt. Islamia Graduate College Kasur Pakistan

---

## [Author Response · Author response to Decision Letter 0]

21 Sep 2023

Journal Requirements:

1)Please ensure that your manuscript meets PLOS ONE's style requirements, including those for file naming.

Response: We have revised the full text with reference to PLOS One's format template requirements.

2)We note that the grant information you provided in the ‘Funding Information’ and ‘Financial Disclosure’ sections do not match.

Response: we have ensured the grant information in the system and match them. 

3)Please state what role the funders took in the study.

Response: We have stated "The funders had no role in study design, data collection and analysis, decision to publish, or preparation of the manuscript”.

4)We note that you have stated that you will provide repository information for your data at acceptance. Should your manuscript be accepted for publication, we will hold it until you provide the relevant accession numbers or DOIs necessary to access your data. If you wish to make changes to your Data Availability statement, please describe these changes in your cover letter and we will update your Data Availability statement to reflect the information you provide.

Response: We have stated that “The minimal data set underlying this study is placed in the public repository, Open Science Framework and can be accessed at the following link: http://www.stats.gov.cn/sj/ndsj/.”

5)We note that all Figures in your submission contain [map/satellite] images which may be copyrighted. All PLOS content is published under the Creative Commons Attribution License (CC BY 4.0), which means that the manuscript, images, and Supporting Information files will be freely available online, and any third party is permitted to access, download, copy, distribute, and use these materials in any way, even commercially, with proper attribution. For these reasons, we cannot publish previously copyrighted maps or satellite images created using proprietary data, such as Google software (Google Maps, Street View, and Earth).

Response: In view of the map copyright issues mentioned by the journal, we have decided to remove the figures from the revised manuscript after careful consideration.

6)Please ensure that you refer to all Figures in your text as, if accepted, production will need this reference to link the reader to the figure.

Response: Not applicable.

7)Please include a separate caption for each figure in your manuscript.

Response: The figures have been removed from the manuscript. This term is not applicable.

Reviewer #1:

(1) Literature Review: This section is well-aligned and presents an effective blend of recent and past studies. Overall, this section is appropriate in my opinion. Kindly add some studies in literature review section.

Response: After reviewing the references given by the reviewer, we selected some literatures relevant to this study and added citations.

Page 2, Line 30-31:

“Many factors including global climate change affect economic development and economic quality [3].

3. Abbass, K., Qasim, M. Z., Song, H., Murshed, M., Mahmood, H., & Younis, I. A review of the global climate change impacts, adaptation, and sustainable mitigation measures. Environmental Science and Pollution Research. 2022; 29(28): 42539-42559. https://doi.org/10.1007/s11356-022-19718-6”

Page 10, Line 217-218: 

“We construct the index system from the five perspectives of innovation, coordination, green, openness and sharing [49-50].

50. Amjad, A., Abbass, K., Hussain, Y., Khan, F., & Sadiq, S. Effects of the green supply chain management practices on firm performance and sustainable development. Environmental Science and Pollution Research. 2022; 29(44): 66622-66639. https://doi.org/10.1007/s11356-022-19954-w”

(2) Study design: This section is well-written and well-explained. Referring to my comments in the Introduction section, try to explain each abbreviation at first and then use its short form.

Response: We checked the full text and added explanations of each abbreviation at first. They were all highlighted in the marked-up copy of our revised manuscript. 

(3) Empirical Results: Results are explained in detail. Must add much more explanations and interpretations for the results, which are not enough. It is suggested to compare the results of the present research with some similar studies which is done before (more justification is needed).

Response: To make the empirical analysis of this research more reliable, we have added hypothesis 4 in the second part “Theoretical Analysis”. 

Page 8, Line 157-159: 

“Both the development of digital economy and consumption growth will impact on high-quality economic development. This impact path includes both consumer demand and consumption supply.”

Page 9, Line 177-180:

“Consequently, hypotheses 3 and 4 were proposed:

H3: Under the regulation of consumption, there is a threshold effect on the impact of the digital economy on high-quality economic development.

H4: Consumption expansion promotes high-quality economic development through the digital economy.”

Also, we revised the models to verify the hypothesis 2 and 4 in the third section “Methodology”.

Page 9, Line 184-193: 

“Firstly, we constructed the benchmark individual fixed-effect model:

〖hqe〗_it=α_0+α_1 〖digit〗_it+〖α_2 〖digit〗_it^2+α〗_3 〖control〗_it+u_i+ε_it (1)

Where 〖hqe〗_it indicates the level of regional high-quality development index of the region i in the period of t. α_0 is the constant. 〖digit〗_it is the digital economy index and 〖digit〗_it^2 is the square of 〖digit〗_it. 〖control〗_it represents the control variables. u_i indicates individual fixed effects and ε_it is the random error. 

The mediational effect model of digit economy is defined as follows.

〖hqe〗_it=β_0+β_1 〖cd〗_it+β_2 〖control〗_it+u_i+ε_it (2)

〖digit〗_it=γ_0+γ_1 〖cd〗_it+γ_2 〖control〗_it+u_i+ε_it (3)

〖hqe〗_it=δ_0+δ_1 〖cd〗_it+〖δ_2 〖digit〗_it+δ〗_3 〖control〗_it+u_i+ε_it (4)”

The results are shown in Table 4.(Page 14)

Mediation effects are further discussed and analyzed in Part IV, section 4.2. 

Page 16, Line 289-321: 

“Mediation effect analysis……

……

It is explained that the consumption supply completely relies on the digit economy to affect the high-quality development.”

In addition, we added discussion to the results of the threshold effect, and analyzed the empirical results of this paper with the existing literature.

Page 21, Line 369-374:

” With the increase in consumer demand, the utility of the digital economy to empower high-quality economic development presents an inverted U-shaped structure. In the benchmark model, we verified the inverted U-shaped structure between the digital economy and the high-quality development. Analyzed from the perspective of consumption expansion, the inverted U-shaped structure may be affected by consumer demand. The validation result of hypothesis 2 is strengthened.”

Page 24, Line 413-424:

” This result differs from the literature [57]. This may be caused by the samples. The literature analyzed based on 30 cities of China……

……

and the development potential of the digital economy is great, which is most effective in enabling the high-quality development of the regional economy.”

(4) Future Directions: Please revise your future research and limitation part into more detail. It would be best if you enhanced your this section.

Response: At the end of the manuscript, we add a paragraph analyzing the limitations and future research directions.

Page 30, Line 561-566: 

“Finally, this study still has limitations and future directions. The spatial econometric models are not utilized to analyze the spatial impact of digital economy and consumption expansion on high-quality development. In addition, how to measure consumption expansion more comprehensively is also worth exploring. Consumption expansion is only one aspect of consumption growth, the consumption quality and its impact on digital economy and high-quality development also deserves further study. “

Reviewer #2:

(1) In order to establish the significance and value of the study, it is necessary to provide a comprehensive rationale for the research, which emphasizes its relevance and unique contributions to the current scholarly discourse. This will strengthen the study's originality and scholarly impact.

Response: By combing through the literatures again, the main contributions of this paper are identified in the last paragraph of Part I. 

Page 5, Line 91-97: 

“From the existing researches, we found that consumption expansion is rarely taken into account in the impact of the digital economy on high-quality development. Will the consumption expansion affect digital economy’s impact on promoting high-quality development? And if it does, what is the relationship between them? What about the impact pattern and effect? These questions are all critical, but the conclusions are yet unclear. Clarifying the role of consumption expansion will help to better achieve high-quality economic development.”

(2) English check/editing: I found minor typos/ grammatical, this study was presented with depraved English language usage.

Response: We have checked on the spelling and grammar of the full text and invited professionals to help and correct the English editing.

(3) The conclusion is sufficiently discussed with some useful policy implications. However, limitations and future directions of research should have a reasonably sized paragraph.

Response: At the end of the manuscript, we add a paragraph analyzing the limitations and future research directions.

Page 30, Line 561-566: 

“Finally, this study still has limitations and future directions. The spatial econometric models are not utilized to analyze the spatial impact of digital economy and consumption expansion on high-quality development. In addition, how to measure consumption expansion more comprehensively is also worth exploring. Consumption expansion is only one aspect of consumption growth, the consumption quality and its impact on digital economy and high-quality development also deserves further study. “

---

## [Decision Letter · Decision Letter 1]

3 Oct 2023

The Impact of Digital Economy on High-quality Economic Development: Research Based on the Consumption Expansion

PONE-D-23-22047R1

Dear Dr. Li,

We’re pleased to inform you that your manuscript has been judged scientifically suitable for publication and will be formally accepted for publication once it meets all outstanding technical requirements.

Kind regards,

Jing Cheng

Academic Editor

PLOS ONE

Additional Editor Comments (optional):

Reviewers' comments:

Reviewer's Responses to Questions

**Comments to the Author**

1. If the authors have adequately addressed your comments raised in a previous round of review and you feel that this manuscript is now acceptable for publication, you may indicate that here to bypass the “Comments to the Author” section, enter your conflict of interest statement in the “Confidential to Editor” section, and submit your "Accept" recommendation.

Reviewer #1: All comments have been addressed

Reviewer #2: All comments have been addressed

2. Is the manuscript technically sound, and do the data support the conclusions?

Reviewer #1: Yes

Reviewer #2: Yes

3. Has the statistical analysis been performed appropriately and rigorously? 

Reviewer #1: Yes

Reviewer #2: Yes

4. Have the authors made all data underlying the findings in their manuscript fully available?

Reviewer #1: Yes

Reviewer #2: Yes

5. Is the manuscript presented in an intelligible fashion and written in standard English?

Reviewer #1: Yes

Reviewer #2: Yes

6. Review Comments to the Author

Reviewer #1: I am glad to review to review this paper " The Impact of Digital Economy on High-quality Economic Development: Research Based on the Consumption Expansion " So , I accept this paper for publication

Reviewer #2: (No Response)

7. PLOS authors have the option to publish the peer review history of their article (what does this mean?). If published, this will include your full peer review and any attached files.

Reviewer #1: **Yes: **KASHIF ABBASS

Reviewer #2: **Yes: **Dr shabbir ahmed department of economics Government islamia graduate college kasur pakistan

---

## [Editor Report · Acceptance letter]

6 Dec 2023

PONE-D-23-22047R1 

The impact of digital economy on high-quality economic development: research based on the consumption expansion 

Dear Dr. Li:

I'm pleased to inform you that your manuscript has been deemed suitable for publication in PLOS ONE. Congratulations! Your manuscript is now with our production department. 

Kind regards, 

on behalf of

Dr. Jing Cheng 

Academic Editor

PLOS ONE